# Cervical Cancer Screening After Menopause

**DOI:** 10.3390/healthcare13101157

**Published:** 2025-05-16

**Authors:** Ho-Jui Tung, Gila Schwarzschild, Nenrot Gopep, Ming-Chin Yeh

**Affiliations:** 1Department of Health Policy and Community Health, Jiann-Ping Hsu College of Public Health, Georgia Southern University, Statesboro, GA 30460, USA; ng06418@georgiasouthern.edu; 2Department of Nutrition and Public Health, Hunter College, City University of New York, New York, NY 10065, USA; gs2024@hunter.cuny.edu (G.S.); myeh@hunter.cuny.edu (M.-C.Y.)

**Keywords:** menopausal transition, middle-aged women, pap smear test, cervical cancer screening

## Abstract

Background: About 14,000 women develop cervical cancer each year in the United States. Human Papillomavirus (HPV) vaccination is an effective primary prevention measure for HPV infections and cervical cancer among adolescents and young adults. For middle-aged and older women, they rely on secondary prevention (i.e., cancer screening) for early detection of cervical cancer. The average age at which women receive a cervical cancer diagnosis is around 50, when most women are in the middle of perimenopause. In this study, we use data from a longitudinal survey to examine whether going through menopause is associated with cervical cancer screening behavior four or eight years later. Methods: Data were taken from 2012, 2016, and 2020 waves of the Health and Retirement Study (HRS), a longitudinal survey of middle-aged and older adults in America. Using the 2012 and 2016 waves as baselines, two four-year (*n* = 1011 and *n* = 1263) and one eight-year (*n* = 823) longitudinal analyses were conducted. The lost follow-ups and those who have had a hysterectomy were excluded. Hierarchical logistic regression models were used to compare women who had gone through menopause to those who were premenopausal or perimenopausal at each of the baselines in terms of their likelihood of having a pap smear test four or eight years later. Results: Results show that the women who had gone through menopause were less likely to have a pap smear test four or eight years later when compared to those who were still premenopausal or perimenopausal at baseline. Women who had gone through menopause at the baseline of 2016 were less likely to have a pap smear test by 2020 (Odds Ratio = 0.76, *p* < 0.05). A similar association was found among women who had gone through menopause at the baseline of 2012 after controlling for their previous pap smear behavior and other covariates. Conclusions: The American Cancer Society and other professional organizations recommend that women have cervical cancer screenings regularly until age 65. Our findings suggest that women seem less likely to have a pap smear test after menopause. More research is needed to have a comprehensive understanding of cervical screening behavior in this age group of women.

## 1. Introduction

Every year, about 14,000 women develop cervical cancer and 4000 die from the disease in the United States [1]. Cervical cancer is largely preventable through routine screening and a follow-up of abnormal test results [2]. Primary prevention of cervical cancer can be achieved with Human Papillomavirus (HPV) vaccination that prevents persistent HPV infections among adolescents and young adults. However, middle-aged and older women rely on secondary prevention measures, such as pap smear tests, for early detection of the cancer [3]. Cervical cancer stands out as the third most prevalent cancer in women worldwide and the fourth most common cause of cancer-related deaths in women [4,5,6,7]. Cervical cancer has been highly associated with persistent infection with human papillomavirus (HPV), particularly the high-risk types [4,8,9], and HPV infection is the most common sexually transmitted disease in the United States and Canada [6,10].

In Western countries like the United States and countries in the European Union, the combination of HPV vaccination and regular screening has significantly reduced cervical cancer incidence [11]. However, this reduction in incidence and mortality from the disease should not mask the fact that considerable ethnic and socio-economic disparity gaps still exist [3]. Factors, such as poverty, low educational level, unemployment, and migrant/refugee or ethnic minority status, are associated with both low health literacy and reduced adherence to cervical cancer screening [3,6]. It is especially disturbing that there was a higher prevalence of cervical cancer among incarcerated women, who were four to five times more likely to report a cervical cancer diagnosis compared to the general population of women [9].

Despite the importance of screening in early detection of the disease among middle-aged women [10], studies have reported that cervical cancer screening participation has decreased among women aged 30 to 64 in the United States over the past few decades [12]. More importantly, a review article points out that more than 50 percent of women who were diagnosed with cervical cancer had inadequate screening, and more than 40 percent had no screening [13].

Around age 40, women begin experiencing significant changes in their reproductive health as they enter the perimenopausal phase, when the function of the ovaries starts declining. This transition continues until menopause, which is marked by the cessation of menstruation for at least one year [14,15]. During this transitional period, women face various health challenges, including an increased likelihood of several cancers [15,16]. Regarding the risk of cervical cancer, a study that combines data from surveys and cancer registries to estimate a hysterectomy-corrected incidence of cervical cancer found that even though the peak of cervical cancer incidence for all US women occurs at ages 40 through 44, the risk escalates with age in those with a cervix until age 70 [10,17]. The American Cancer Society, American College of Obstetricians and Gynecologists, and U.S. Preventive Services Task Force recommend regular screening for cervical cancer between the ages of 25 and 65 [18,19].

However, numerous studies have revealed concerning trends in postmenopausal screening behaviors and outcomes. Using insurance claims, one study found that only 37 percent of women with employer-based insurance were appropriately screened between the ages of 54 and 64, and 12 to 18 percent of women ages 45 to 65 reported no cervical cancer screening for more than 5 years [20]. On the other hand, there seems to be a misperception linking sexual activities with cervical cancer. A large sample study using data from the UK Biobank on women ages 37 through 70 found that women who had sex exclusively with women or who never had sex had lower odds of ever undergoing cervical cancer screening compared with heterosexual women. Plus, among women with no history of sex with either women or men, 45.7 percent reported never having a cervical cancer screening [10,21]. In another study conducted in Hong Kong on women aged 50 or over, 66 percent of study respondents believed that screening was not necessary after menopause [22]. This misconception has a particularly troubling consequence, given that postmenopausal women are especially vulnerable to cervical cancer [14]. Evidence shows that, among older women, regular screening could reduce the incidence and mortality of cervical cancer substantially [23]. Since menopause marks the end of the reproductive life of a woman, could the aforementioned misconception be associated with women’s decisions to undergo pap smear testing? In this study, we investigate whether there is an association between menopause and cervical cancer screening. Using data from a longitudinal survey, we examine whether women who had gone through menopausal transition would be less likely to have pap smear tests four and eight years later.

## 2. Methods

### 2.1. Data and Samples

Launched in 1992, the Health and Retirement Study (HRS) is a biennial panel-design longitudinal survey on a nationally representative sample of adults over 51 in the United States [24]. However, some HRS participants were 50 years old or younger when they were first interviewed. The observational unit of the survey is the household financial unit. If a sampled participant’s housing unit contains more than one age-eligible person, one of these persons will be randomly selected to be included in the survey. Plus, if an age-eligible person has a spouse, the spouse is automatically included in the study, even if he or she is 50 or younger [24].

The HRS does not administer all survey questions biennially, and the menopause stage question is one of these partial interview questions. A great majority of HRS respondents had already gone through menopause long before their first interview. Only a fraction of the HRS participants were asked about their menopause stage: “Regarding menopause, do you think you are without a sign, just beginning, in the middle, near the end, or all through?”. Based on their previous response to this question, those who responded with “all through” will not be asked about their menopause stage in the current wave. Since a menopausal transition could take longer than two years, our longitudinal analyses would focus on the 2012, 2016, and 2020 waves of HRS to examine the association between menopausal transition and having a pap smear test over four-year and eight-year follow-up periods. The HRS participants who reported having had a hysterectomy at the baseline were excluded from the study since they were no longer at risk of cervical cancer [25]. That left 1011 cases (from the baseline of 2012 to the follow-up of 2016), 1263 cases (from the baseline of 2016 to the follow-up of 2020), and 823 cases (from the baseline of 2012 to the follow-up of 2020) for the current analysis.

### 2.2. Measures

The dependent variable, having a pap smear test at the follow-ups, was measured by the 2016- and 2020-wave survey item, “In the last two years, have you had a pap smear test?”. A “yes” was coded 1 (having a pap smear in the last 2 years), and all other responses were coded 0. For the analysis from the baseline of 2012 through 2020, respondents who answered “yes” in either 2016 or 2020 follow-ups were counted as having had a pap smear test (coded 1).

The independent variable, the menopause stage, was measured by the question, “Regarding menopause, do you think you are without a sign, just beginning, in the middle, near the end, or all through?”. Those who answered the question with responses of “near the end” and “all through” were combined and classified as having gone through a menopausal transition four years later (coded 1). The other three responses (e.g., “without a sign”, “just beginning”, and “in the middle”) were collapsed and classified as premenopausal or perimenopausal. This dichotomous variable was used in the logistic regression models.

Other covariates include age, race, educational attainment, marital status at the baseline, having a usual care place, and having had a pap smear test before. Our logistic regression models treated age (chronological age) at baseline and educational attainment (measured as years of schooling) as continuous measures. The race variable was dichotomized into Black (=1) and all others (=0). Marital status at baseline was a dichotomous measure (1 = married and 0 = otherwise). “Having a usual place of care” (1 = yes; 0 = otherwise) was also included in predicting the dependent variable. Finally, previous pap smear experience was measured by asking respondents if they had had a pap smear test in the past two years (1 = yes; 0 = otherwise) at the baselines of 2012 and 2016. This variable was added to the hierarchical logistic regression models on top of all other covariates (Mode 2) to examine the changes in odds ratio estimates of the independent variables across different hierarchical logistic regression models.

### 2.3. Analysis

Hierarchical logistic regressions predicting the likelihood of a pap smear test at the follow-ups (baseline of 2012 to 2016, baseline of 2016 to 2020, and baseline of 2012 through 2020) were used to evaluate if menopausal transition is associated with the uptake of cervical cancer screening across different study periods and durations after menopause. First, our primary independent variable (the dichotomous menopause stage) and covariates were entered into the logistic regression (Model 1) to predict the dependent variable. In Model 2, previous pap smear experience was added to the model to predict the likelihood of having a pap smear at the follow-ups. By doing so, an attempt is made to examine if the significance level of the primary independent variable and the other covariates might be altered in predicting the likelihood of having a pap smear at the follow-ups. Lastly, using respondents from the baseline sample of 2012, who were re-interviewed in both the 2016 and 2020 follow-ups (*n* = 823), a third logistic regression, predicting if they had a pap smear test in either 2016 or 2020 (coded 1), was conducted to examine the association between menopausal transition and the uptake of pap smear tests over 8 years.

## 3. Results

Descriptive statistics of the two baseline samples are presented in Table 1. After excluding respondents who had reported previously having a hysterectomy, the baseline of 2012 had a sample size of 1011 respondents (mean age = 49.9); most (87.6 percent) were 45 years or older in 2012. In terms of their menopausal stages, about 30.8 percent (311 out of 1011) of them reported that they were either “near the end” or “all through”. The same procedure was applied to handle the baseline sample of 2016. About 90.1 percent of the 1263 participants (mean age = 50.2) were 45 years or older in 2016, and 34.4 percent (434 out of 1263) reported that they were either “near the end” or “all through”.

Table 2 presents the logistic regression predicting the odds of having a pap smear test in 2016 for the baseline sample of 2012. From Model 1, we can see that female respondents who were transitioning to menopause in 2012 were 32 percent less likely to have a pap smear in 2016 (Odds Ratio, OR = 0.68, *p* < 0.05) when compared to those who were still premenopausal in 2012. Married women and those who had a usual place of care were more likely to have pap smear screening in the 2016 follow-up. When the respondents’ previous pap smear screening was included in the models, the significance of having usual place of care and marital status disappeared (Model 2), implying that their significant associations with the dependent variable in Model 1 can be explained away by the newly introduced predictor, previous pap smear tests. However, a borderline significance (OR = 0.71, *p* = 0.05) was found for our primary independent variable in predicting the odds of having a pap smear test in 2016, meaning that menopausal stage was still somewhat significant in predicting the dependent variable. A similar association pattern was revealed when the 2016 baseline sample was analyzed. From Table 3, we can see that female respondents transitioning to menopause in 2016 were 26 percent less likely to have a pap smear in 2020 (OR = 0.74, *p* < 0.05) compared to those who were still premenopausal or perimenopausal. We also found that African American women were 1.72 times (*p* < 0.001) more likely to have a pap smear test when compared to other racial/ethnic groups. When previous pap smear screening was controlled for, women who were transitioning to menopause in 2016 were still less likely (OR = 0.76, *p* < 0.05) to have a pap smear test in 2020 (Model 2 of Table 3), while those who were married and/or African American were more likely to have a pap smear test in the 2020 follow-up.

Table 4 presents a longer-term observation of the association between the HRS respondents’ menopause stages in 2012 and pap smear screenings over the following eight years (the 2016 and 2020 follow-ups). As we can see, before controlling for their previous pap smear screening, women who were transitioning to menopause were less likely (OR = 0.59, *p* < 0.05) to have a pap smear test in either 2016 or 2020, while women who were married and who had a usual place of care were more likely to have one in the follow-up years. When the previous pap smear screening behavior was controlled for, the significant effect of the menopausal transition in predicting a pap smear test remained (OR = 0.64, *p* < 0.05). At the same time, the significance of the other two predictors disappeared.

## 4. Discussion

Most cervical cancer cases are diagnosed in middle-aged and older women who rely on secondary prevention for early detection of the cancer [11]. Lack of screening could lead to a higher risk of advanced stages of cervical cancer and mortality at diagnosis [26,27]. In this preliminary study, our findings suggest that, among the HRS participants who had just undergone menopausal transition, they seemed less likely to have a pap smear test four or eight years later than those who were still premenopausal or perimenopausal at baseline. The significant difference remained after controlling for previous pap smear test behavior and other covariates.

It is unclear why the participants in our study had lower cervical screening uptakes after menopause, and we consider these findings as preliminary. However, it is speculated that a less frequent cervical screening among postmenopausal women might have something to do with the fact that, different from other secondary preventive measures for cancers, cervical screening involves an intrusive device, anxiety, and stigma associated with the etiology and risk factors of cervical cancer (e.g., sexually transmitted infections of HPV). As mentioned above, Saunders and her colleagues, using data from the UK Biobank, found that women who never had sex had lower odds of ever undergoing cervical cancer screening when compared with heterosexual women [21]. This implies that, to a great extent, cervical cancer is considered by them a sexually transmitted disease. By the same token, women after menopause may become less sexually active due to menopause-related changes [28]. Consequently, they may think that they have a lower susceptibility to cervical cancer and cervical screening is not necessarily needed.

On the other hand, having a pap smear test is an unpleasant experience for many women. A survey of a group of African American women has found that the pain incurred by a pap smear screening discouraged middle-aged African American women from adhering to the screening [29]. In a review of eighteen qualitative studies published between 2012 and 2021 about women’s critical experiences with going through a pap smear test, Arrivillaga and colleagues identified four themes reported in these studies: *fear and embarrassment* during the procedure, *pain and discomfort* due to the speculum, *distress* due to the implications of test results, and *barriers to health services* [30]. There are also reports about the anxiety and adverse responses caused by an abnormal screening result [31,32]. An abnormal result is especially troublesome due to the link between abnormal results and sexually transmitted HPV infections. Communication strategies addressing the stigma associated with STIs are needed for public health campaigns in promoting cervical cancer screening.

Our study has several limitations. First, as a secondary analysis of survey data, the dependent and independent variables and the covariates were all self-reported measures. Plus, the inclusion and exclusion of participants indeed lacked clinically defined criteria to verify their menopausal status objectively. However, women are keenly aware of the menstrual changes. It is argued that this self-reported menopause stage can be a good proxy measure for where they stand regarding their menopausal transition. Second, our dependent variable only captured pap smear uptakes within two years, while the follow-up period extended to four or eight years. This may have resulted in an underestimate of screenings in the dependent variable. However, it is assumed that this mismatch between the measurement timeline and the duration of follow-up should impact women in different menopausal stages equally. Third, the samples used in this study do not represent all women undergoing menopausal transition in the United States. As explained in the Section 2, only a fraction of the female HRS participants were interviewed for their menopausal transition.

Lastly, there is always a possibility that some unmeasured confounders could explain what we found in this study. However, significant differences in the uptake of pap smear tests among postmenopausal women were found across different study periods and durations of follow-up. Yet, the significant associations remained after the potent predictor, their past testing experience, had been controlled for. Based on our preliminary findings, more research should be conducted to clarify this critical public health issue.

## 5. Conclusions

In this secondary analysis of survey data, a preliminary finding on lower uptake of cervical cancer screening was found among women who had gone through their menopausal transition. For middle-aged women, cervical cancer is largely preventable through early detection. However, evidence has shown that cervical screening participation has been declining among this age group of women. Studies with more objective measures and sophisticated designs are needed to identify the underlying mechanisms leading to lower uptake of cervical screening among postmenopausal women, so that tailored interventions can be designed to address this critical public health issue.

## Figures and Tables

**Table 1 healthcare-13-01157-t001:** Selected characteristics of female HRS participants experiencing menopausal transition at the baselines of 2012 and 2016.

Predictors	Baseline of 2012 Predicting Follow-Up in 2016 (*n* = 1011)	Baseline of 2016 Predicting Follow-Up in 2020 (*n* = 1263)
Menopause stage (menopause = 1)	311 (30.8)	434 (34.4)
Mean Age in 2012	49.9 (4.76)	---
Mean Age in 2016	---	50.2(4.55)
Mean Schooling Years	13.0 (3.09)	13.1 (3.39)
African Americans (yes)	247 (24.4)	324 (25.7)
Marital status in 2012 (married)	686 (67.9)	---
Marital Status in 2016 (married)	---	743 (58.8)
Had usual place of care in 2012 (yes)	836 (82.7)	---
Had usual place of care in 2016 (yes)	---	1052 (83.3)
Had a previous pap smear in the last 2 years in 2012 (yes)	721 (71.3)	---
Had a previous pap smear in the last 2 years in 2016 (yes)	---	934 (74.0)
Having a pap smear at follow-up in 2016 (yes)	723 (71.5)	---
Having a pap smear at follow-up in 2020 (yes)	---	794 (62.9)

Note: 1. For categorical variables, the number of cases and percentages (parentheses) are shown; 2. For continuous variables, means and standard deviations (parentheses) are shown.

**Table 2 healthcare-13-01157-t002:** Odds ratios of having a pap smear test in 2016 among female HRS participants from the baseline of 2012 (*n* = 1011).

Predictors	Odds Ratio (95% Confidence Intervals)	
	Model 1	Model 2
Menopause stage (“near the end” or “all through” = 1)	0.68 (0.49, 0.93) *	0.71 (0.51, 1.00) ^α^
Age in 2012	1.02 (0.98, 1.05)	1.02 (0.98, 1.05)
Schooling Years	1.01 (0.97, 1.06)	1.00 (0.95, 1.05)
African American (yes)	1.33(0.95, 1.86)	1.15 (0.80, 1.65)
Marital status in 2012 (yes)	1.54 (1.15, 2.08) **	1.34 (0.98, 1.85)
Had usual place of care (yes)	1.48 (1.03, 2.12) *	1.11 (0.75, 1.65)
Had a pap smear in 2012 (yes)	---	5.52 (4.07, 7.52) ***
−2 Log Likelihood/degrees of freedom	1185.92/6	1061.89/7

Note: 1. * *p* < 0.05, ** *p* < 0.01, *** *p* < 0.001, ^α^ *p* = 0.05 (borderline significance); 2. The difference of −2 Log Likelihood between Models 1 and 2 is 124.03. With 1 degree of freedom (7-6), it implies a significant improvement in model fit for Model 2.

**Table 3 healthcare-13-01157-t003:** Odds ratios of having a pap smear test in 2020 among HRS female participants, from the baseline of 2016 (*n* = 1263).

Predictors	Odds Ratio (95% Confidence Intervals)	
	Model 1	Model 2
Menopause stage (“near the end” or “all through” = 1)	0.74 (0.58, 0.96) *	0.76 (0.58, 0.99) *
Age in 2016	1.01 (0.98, 1.04)	1.01 (0.98, 1.04)
Schooling Years	1.02 (0.96, 1.06)	1.01 (0.97, 1.04)
African American (yes)	1.72 (1.29, 2.28) ***	1.63 (1.21, 2.19) **
Marital status in 2016 (yes)	1.39 (1.09, 1.78) **	1.29 (1.00, 1.67) *
Had usual place of care (yes)	1.40 (1.03, 1.91) *	1.21 (0.87, 1.67)
Had a previous pap smear at baseline of 2016 (yes)	---	3.56 (2.73, 4.65) ***
−2 Log likelihood/Degrees of Freedom	1635.46/6	1545.96/7

Note: 1. * *p* < 0.05, ** *p* < 0.01, *** *p* < 0.001; 2. The difference of −2 Log Likelihood between Models 1 and 2 is 89.50. With 1 degree of freedom (7-6), it implies a significant improvement in model fit for Model 2.

**Table 4 healthcare-13-01157-t004:** Odds ratios of having a pap smear test in either 2016 or 2020 among HRS female participants from the baseline of 2012 (*n* = 823).

Predictors	Odds Ratio (95% Confidence Intervals)	
	Model 1	Model 2
Menopause stage (“near the end” or “all through” = 1)	0.59 (0.39, 0.90) *	0.64 (0.41, 0.99) *
Age in 2012	1.00 (0.95, 1.04)	0.99 (0.94, 1.04)
Schooling Years	1.01 (0.95, 1.07)	1.00 (0.94, 1.07)
African American (yes)	1.29 (0.82, 2.04)	1.08 (0.67, 1.75)
Marital status in 2016 (yes)	1.61 (1.08, 2.38) *	1.39 (0.91, 2.11)
Had usual place of care (yes)	1.64 (1.02, 2.62) *	1.23 (0.75, 2.04)
Had a previous pap smear at baseline of 2012 (yes)	---	5.79 (3.88, 8.63) ***
−2 Log likelihood/Degrees of Freedom	720.99/6	644.46/7

Note: 1. * *p* < 0.05, *** *p* < 0.001; 2. The difference of −2 Log Likelihood between Models 1 and 2 is 76.53. With 1 degree of freedom (7-6), it implies a significant improvement in model fit for Model 2.

## Data Availability

The data used in this study were downloaded from a publicly accessible website https://hrs.isr.umich.edu/about, accessed on 10 May 2025.

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
