# Peer review of "Cervical Cancer Screening After Menopause"

_healthcare, 2025, doi:10.3390/healthcare13101157_

Round 1

Reviewer 1 Report

Comments and Suggestions for Authors

Thank you so much for the opportunity to review your paper. Below, few comments you might consider or clarify:

 The manuscript groups responses “near the end” and “all through” as indicators of having undergone menopausal transition. This categorization is not fully justified. It would be useful to clarify how these self-reported categories capture the biological and clinical aspects of menopause and whether this grouping might lead to misclassification.

The dependent variable is defined as having had a pap smear in the past two years, yet follow-up periods extend to four or eight years. This mismatch could result in under- or overestimating screening behavior over the entire follow-up period. More discussion or sensitivity analyses are needed to address this inconsistency.

 In the hierarchical logistic models, previous pap smear screening is a very strong predictor, causing shifts in the significance of other covariates. This raises questions about whether the independent effect of menopausal status is being accurately isolated. A more detailed explanation of the rationale for controlling for prior screening—and its potential impact on interpreting the effect of menopause—would improve clarity.

 The odds ratios reported for menopausal transition (e.g., 0.68, 0.74, 0.59 in the different models) suggest a moderate association with screening behavior. However, the change in significance after adjusting for prior screening behavior (e.g., borderline significance in some models) is not fully explored. The discussion should more carefully interpret these changes and acknowledge the limitations of drawing causal inferences from these observational data.

While the analyses indicate lower odds of screening among women classified as postmenopausal, the possibility of unmeasured confounders (beyond those included) could lead to either an over- or underestimation of the true association. This concern should be more clearly acknowledged in both the results and discussion sections.

  - The tables present odds ratios with 95% confidence intervals, but additional details—such as the exact p-values and test statistics (e.g., Chi-square values or -2 Log Likelihood differences)—would help readers assess the robustness of the models.
  - In several tables, the changes between Model 1 and Model 2 (after including previous pap smear behavior) are not fully explained. Consider adding footnotes or expanding the text to clarify why certain predictors lose significance.

The text generally describes lower screening uptake among women post-menopause; however, the narrative should more explicitly acknowledge the borderline significance in some models. For example, in Table 2, the adjusted odds ratio (0.71, p=0.05) is on the threshold of significance, which calls for a more cautious interpretation in the discussion.

The discussion suggests that women “lower their guard” after menopause. Given the observational nature of the study and the limitations noted (such as potential misclassification and unmeasured confounders), the language should be more tentative and avoid implying causality without further supporting evidence.

 Some sections—particularly those describing the hierarchical logistic regression models—would benefit from clearer language. It is important to describe the stepwise inclusion of covariates and how the interpretation of the menopausal transition effect changes with these adjustments.

Looking forward seeing your revised paper

Best wishes

Author Response

Response to Reviewer 1 Comments

Point-by-point response to comments and suggestions for authors:

Comment 1: The manuscript groups responses “near the end” and “all through” as indicators of having undergone menopausal transition. This categorization is not fully justified. It would be useful to clarify how these self-reported categories capture the biological and clinical aspects of menopause and whether this grouping might lead to misclassification.

Response 1: We agree. Our study is a secondary data analysis; this self-reported question regarding a participant’s menopause stage (the primary independent variable, IV) was the only information we could obtain. It is argued that women are keenly aware of their menstrual changes. This self-reported measure can be a good proxy measure for where they stand regarding their menopausal transition. To accommodate this limitation, we decided to examine the association between our IV and having a pap smear test (DV) over four and eight years, so that, for those whose menopause stage was “near the end,” they would be likely to have gone through the menopausal transition by then. We have emphasized this limitation in the Discussion section (starting from line 236)

Comment 2: The dependent variable is defined as having had a pap smear in the past two years, yet follow-up periods extend to four or eight years. This mismatch could result in under- or overestimating screening behavior over the entire follow-up period. More discussion or sensitivity analyses are needed to address this inconsistency.

Response 2: We agree. Our decision to analyze the data this way was indeed a compromise. First, the mismatch in timing for the IV and DV is because of the Health and Retirement Study (HRS) survey timeline. The HRS does not administer all survey questions biennially. HRS randomly selects half of its sample for the enhanced interview and the other half completes only the core interview, usually by telephone. Our rationale was that, even though the DV question asks about participants’ pap smear tests in the past two years, due to the random selection, this mismatch should impact the two categories of the IV (the premenopausal/perimenopausal and the postmenopausal) equally.     

Comment 3: In the hierarchical logistic models, previous pap smear screening is a very strong predictor, causing shifts in the significance of other covariates. This raises questions about whether the independent effect of menopausal status is being accurately isolated. A more detailed explanation of the rationale for controlling for prior screening—and its potential impact on interpreting the effect of menopause—would improve clarity.

Response 3: Thank you for pointing out this concern. As a retrospective study using secondary data, we did not have all the covariates we wanted to control for. So, instead of trying to establish a comprehensive predictive model for cervical screening, our analyses focused on whether women, when they had gone through menopausal transition, were less likely to have a Pap smear test. By including the most potent predictor (previous pap smear test) in the hierarchical logistic regressions, and if there is still a significant association between the IV and DV, it is likely that the association is not spurious. However, it is still possible that some unmeasured factors might be associated with the dependent variable. In the Discussion section, we addressed this issue as a study limitation (starting from Line 236) to remind readers to treat the findings as preliminary. We have revised our interpretation of findings with more qualifications (starting from Line 206)

Comment 4: The odds ratios reported for menopausal transition (e.g., 0.68, 0.74, 0.59 in the different models) suggest a moderate association with screening behavior. However, the change in significance after adjusting for prior screening behavior (e.g., borderline significance in some models) is not fully explored. The discussion should more carefully interpret these changes and acknowledge the limitations of drawing causal inferences from these observational data.

Response 4: We agree. There are not many studies on the association between menopause and cervical cancer screening so far, and more research is needed to confirm the association identified in this study. We have added more elaborations on the interpretation of odds ratio and changes between Model 1 and Model 2 (starting from Line 176).

Comment 5: The tables present odds ratios with 95% confidence intervals, but additional details—such as the exact p-values and test statistics (e.g., Chi-square values or -2 Log Likelihood differences)—would help readers assess the robustness of the models.

Response 5: Thank you for the comment. In Tables 2, 3, and 4, we reported odds ratio estimates and their 95% Confidence Intervals. As suggested, we have added the -2 Log Likelihood differences at the bottom of Table 2 (Line 262), Table 3 (Line 365), and Table 4. (Line 368).

Comment 6: In several tables, the changes between Model 1 and Model 2 (after including previous pap smear behavior) are not fully explained. Consider adding footnotes or expanding the text to clarify why certain predictors lose significance.

Response 6: Thank you for the comment. As suggested, we have added paragraphs in the Results section to explain the odds ratio changes between Model 1 and Model 2 (starting from Line 176).

Comment 7: The text generally describes lower screening uptake among women post-menopause; however, the narrative should more explicitly acknowledge the borderline significance in some models. For example, in Table 2, the adjusted odds ratio (0.71, p=0.05) is on the threshold of significance, which calls for a more cautious interpretation in the discussion.

Response 7: Thank you for pointing out the concern. In Table 2, one of the odds ratio estimates for the IV is at a borderline significance level (p=0.05). We marked the odds ratio with a different symbol and put a footnote at the bottom of Table 2 to remind readers of this weakness. We have added sentences in the Results section (Line 182) to remind readers of this borderline significance.

Comment 8: The discussion suggests that women “lower their guard” after menopause. Given the observational nature of the study and the limitations noted (such as potential misclassification and unmeasured confounders), the language should be more tentative and avoid implying causality without further supporting evidence.

Response 8: Thank you for pointing out the concern. We have deleted all the sentences with “lower their guard” in the abstract and main text body.

Comment 9: Some sections—particularly those describing the hierarchical logistic regression models—would benefit from clearer language. It is important to describe the stepwise inclusion of covariates and how the interpretation of the menopausal transition effect changes with these adjustments.

Response 9: Thank you for pointing out the concern. We have revised the wording in the Results section (starting from Line 155) to explain in more detail regarding the hierarchical logistic regression models.

Reviewer 2 Report

Comments and Suggestions for Authors

This manuscript retrospectively evaluates the association between menopausal transition and cervical cancer screening behavior using data from the nationally representative Health and Retirement Study (HRS). The authors conducted a longitudinal analysis over two 4-year periods and one 8-year period, offering valuable insight into temporal changes in screening behavior across the menopausal transition. The study addresses a public health issue of significant relevance—declining cervical cancer screening rates among postmenopausal women—which has important implications for cancer prevention and health equity. This underscores the need for targeted public health interventions and educational messaging tailored specifically to the postmenopausal population.

There are several major and minor issues addressed below that require clarification before the consideration of acceptance.

  • The inclusion and exclusion criteria for study participants need to be more clearly defined.
    • It is not stated whether women with a history of abnormal Pap smears (e.g., LSIL, HSIL, or AGC), prior cervical procedures (such as LEEP or CKC), or known HPV infection/HPV status were excluded from the analysis.
    • Clarifying whether participants all had normal baseline screening results is essential to contextualize the findings.
    • Additionally, information regarding prior HPV status is not mentioned but may be relevant as a confounder influencing both screening frequency and clinical recommendations.
    • Include a flow diagram summarizing inclusion/exclusion criteria.
  • Menopausal status is based on self-reported survey responses. This introduces the possibility of recall bias and misclassification. Menopausal status, in particular, is based on a subjective self-assessment and may not correspond to clinical definitions, such as those based on hormonal levels or amenorrhea duration.
    • It would strengthen the manuscript to acknowledge this limitation explicitly in the Discussion.
    • Including a table of coding schemes (e.g., how menopausal status and screening behavior were categorized) would improve transparency.
  • The study design defines the outcome as having received a Pap smear “within the last two years”, with follow-up periods occurring “4 and 8 years after baseline”. However, current U.S. cervical cancer screening guidelines (e.g., ASCCP) recommend routine Pap testing every three to five years, depending on age and screening method (cytology only versus HPV co-testing).
    • The authors should clarify the rationale for using a two-year interval to define screening status and for selecting follow-up timepoints at 4 and 8 years, rather than aligning more closely with five- and ten-year intervals. This mismatch between clinical recommendations and analytic timeframes raises concerns about potential underestimation of screening adherence. It is important for the authors to explain how this temporal discrepancy was addressed analytically and to discuss its possible impact on the validity and interpretation of the results.
  • Several important covariates appear to be missing from the analysis. Factors such as hormone replacement therapy (HRT) use, comorbidities, healthcare access frequency, and sexual activity, could significantly affect cervical cancer screening behavior and should at least be discussed as limitations. Without accounting for these potential confounders, the observed associations between menopausal status and screening behavior may be influenced by unmeasured variables.
  • The authors briefly mention that African American women were more likely to undergo screening, what about other racial minority?
    • Reporting outcomes across other racial and ethnic groups, such as Hispanic women, could provide additional insight and enhance the manuscript’s relevance to health equity.
  • Strengthen practical recommendations in the Conclusion—how should providers intervene? Should guidelines emphasize risk-based screening over age-based?
  • Table descriptions could also be improved—some are repetitive
  • The terms "perimenopausal" and "premenopausal" are used inconsistently; these should be clearly defined and used accurately throughout the manuscript.

Author Response

Responses to Reviewer 2 Comments

General comments: This manuscript retrospectively evaluates the association between menopausal transition and cervical cancer screening behavior using data from the nationally representative Health and Retirement Study (HRS). The authors conducted a longitudinal analysis over two 4-year periods and one 8-year period, offering valuable insight into temporal changes in screening behavior across the menopausal transition. The study addresses a public health issue of significant relevance—declining cervical cancer screening rates among postmenopausal women—which has important implications for cancer prevention and health equity. This underscores the need for targeted public health interventions and educational messaging tailored specifically to the postmenopausal population.

Response to general comments: We genuinely appreciate the reviewer’s time in evaluating our manuscript.

Point-by-point response to comments and suggestions for authors:

Comment 1: The inclusion and exclusion criteria for study participants need to be more clearly defined. It is not stated whether women with a history of abnormal Pap smears (e.g., LSIL, HSIL, or AGC), prior cervical procedures (such as LEEP or CKC), or known HPV infection/HPV status were excluded from the analysis. Clarifying whether participants all had normal baseline screening results is essential to contextualize the findings. Additionally, information regarding prior HPV status is not mentioned but may be relevant as a confounder influencing both screening frequency and clinical recommendations.

Response 1: Thank you for the comments. The inclusion and exclusion of our study participants indeed lacked clinically defined criteria. We took data from a national longitudinal survey, and all the variables in this study were self-reported. It would be great to have these clinical results to corroborate and contextualize study findings. However, we did not have them. It's also worth noting that women with positive HPV or LSIL results still require more frequent pap smear follow-ups compared to those with negative results. This limitation would not contradict our analysis findings. Plus, our study purpose was more focused on whether women, when they had gone through menopausal transition, were less likely to have a Pap smear test. We are aware of these limitations, and we have added sentences to remind readers to treat the findings as preliminary (starting from line 236).

Comment 2: Include a flow diagram summarizing inclusion/exclusion criteria.

Response 2: A flow diagram of how we selected the participants is presented below.

Comments 3: Menopausal status is based on self-reported survey responses. This introduces the possibility of recall bias and misclassification. Menopausal status, in particular, is based on a subjective self-assessment and may not correspond to clinical definitions, such as those based on hormonal levels or amenorrhea duration. It would strengthen the manuscript to acknowledge this limitation explicitly in the Discussion.

Response 3: We agree. As mentioned above, the study's dependent and independent variables (DV and IV) were self-reported. We did not have clinical results to verify the DV and IV. We have acknowledged this limitation and emphasized that our findings are preliminary and that more research is needed to confirm the association between the IV and DV (starting from line 258).    

Comments 4: Including a table of coding schemes (e.g., how menopausal status and screening behavior were categorized) would improve transparency.

Response 4: Thank you for pointing out this concern. We have revised the methods section to improve the description of how the independent and dependent variables were measured in the study. (starting from line 143).

Comments 5: The study design defines the outcome as having received a Pap smear “within the last two years”, with follow-up periods occurring “4 and 8 years after baseline”. However, current U.S. cervical cancer screening guidelines (e.g., ASCCP) recommend routine Pap testing every three to five years, depending on age and screening method (cytology only versus HPV co-testing). The authors should clarify the rationale for using a two-year interval to define screening status and for selecting follow-up timepoints at 4 and 8 years, rather than aligning more closely with five- and ten-year intervals. This mismatch between clinical recommendations and analytic timeframes raises concerns about potential underestimation of screening adherence. It is important for the authors to explain how this temporal discrepancy was addressed analytically and to discuss its possible impact on the validity and interpretation of the results.

Response 5: We agree. Our decision to analyze the data this way was indeed a compromise. First, the mismatch in timing for the IV and DV is because of the Health and Retirement Study (HRS) survey timeline. To save money, HRS does not administer all survey questions biennially. HRS randomly selects half of its sample for the enhanced interview and the other half completes only the core interview, usually by telephone. Our DV only captured pap smear receipts within two years, while the follow-up period extended to four or eight years. This methodological limitation may have resulted in some screenings being unrecorded. However, our rationale was that this underestimation should impact the two IV categories (women before and after menopause) equally. We have added these qualifications into the Discussion section (starting from line 236).

Comments 6: Several important covariates appear to be missing from the analysis. Factors such as hormone replacement therapy (HRT) use, comorbidities, healthcare access frequency, and sexual activity, could significantly affect cervical cancer screening behavior and should at least be discussed as limitations. Without accounting for these potential confounders, the observed associations between menopausal status and screening behavior may be influenced by unmeasured variables.

Response 6: Thank you for the comment. Unfortunately, we did not have these covariates from the HRS. So, instead of establishing a comprehensive predictive model for the uptake of a pap smear test, this study focused on whether women, when they had gone through menopausal transition, were less likely to have a Pap smear test. By including the most potent predictor, previous pap smear test experience, and if there is still a significant association between the IV and DV, it is likely that the association is not spurious. We agree that it is still possible that some unmeasured confounders might be at play. In the Discussion section, we addressed this issue as a study limitation (starting from line 236 and Line 258) to remind readers to consider the findings as preliminary.

Comments 7: The authors briefly mention that African American women were more likely to undergo screening, what about other racial minority? Reporting outcomes across other racial and ethnic groups, such as Hispanic women, could provide additional insight and enhance the manuscript’s relevance to health equity.

Response 7: Thank you for the comment. In the data released by the HRS, there were only four categories in the racial/ethnic variable: white, African American, Hispanic, and all others. We have analyzed the data with different racial/ethnic combinations and found that the only significant comparison across several combinations was the one between African Americans and all others. The literature also suggests that African American women had lower cervical cancer screening rates compared to other races. That’s why we made the race a dichotomous variable (African American versus otherwise). We agree that racial health equity is a significant public health priority; however, this study was more focused on whether women, when they had gone through menopausal transition, were less likely to have a Pap smear test. That’s why we did not have much discussion on that.

Comments 8: Strengthen practical recommendations in the Conclusion—how should providers intervene? Should guidelines emphasize risk-based screening over age-based?

Response 8: Thank you for your valuable suggestion. An independent section for the Conclusion has been added at the end of the manuscript. (Starting from Line 258)

Comments 9: Table descriptions could also be improved—some are repetitive.

Response 9: Thank you for pointing out. Table descriptions were misplaced when it was converted to the Journal format by the journal editors. Corrections have been made (Table 1 through Table 4)

Comments 10: The terms "perimenopausal" and "premenopausal" are used inconsistently; these should be clearly defined and used accurately throughout the manuscript.

Response 10: Thank you for the comment. We have revised and used these terms consistently. Most of the participants in our study were in the perimenopausal (the transition phase to menopause) stage at the two baselines. About 87.6 percent were 45 years or older in the 2012 sample, and about 90.1 percent were 45 years or older in the 2016 sample. However, a few participants were premenopausal, referring to the stage before the onset of perimenopause.

Round 2

Reviewer 1 Report

Comments and Suggestions for Authors

Thank you for addressing the comments 

Reviewer 2 Report

Comments and Suggestions for Authors

This revised manuscript is substantially improved, and the authors have adequately addressed the concerns raised in the previous review. The authors clearly acknowledge the limitations inherent to the use of survey data, which strengthens the manuscript by helping readers better understand the study’s design compromises. Their discussion highlights the importance of future research to explore unmeasured covariates not captured in this analysis. I have no further comments at this time.